Automatic spread factor and position definition for UAV gateway through computational intelligence approach to maximize signal-to-noise ratio in wooded environments

Cardoso Caio M. M. caio.cardoso@itec.ufpa.br
Macedo Alex S.
Fernandes Filipe C.
http://orcid.org/0000-0001-9033-6410 Cruz Hugo A. O.
http://orcid.org/0000-0003-0487-0049 Barros Fabrício J. B.
http://orcid.org/0000-0003-3514-0401 de Araújo Jasmine P. L.
Postgraduate Program in Electrical Engineering, Universidade Federal do Pará , Belém, Pará , Brazil
See Chan Hwang
Electronic publication date: 2024 Sep 27
Publication date: 2024
Volume: 10
Electronic Location ID: e2237
Received 2024 Feb 5; Accepted 2024 Jul 14
Copyright: © 2024 Cardoso et al.
Copyright year: 2024
Copyright holder: Cardoso et al.
License: This is an open access article distributed under the terms of the Creative Commons Attribution License, which permits unrestricted use, distribution, reproduction and adaptation in any medium and for any purpose provided that it is properly attributed. For attribution, the original author(s), title, publication source (PeerJ Computer Science) and either DOI or URL of the article must be cited.
License URL: https://creativecommons.org/licenses/by/4.0/

Keywords: LoRa, UAV, Optimization, Grey-Wolf Optimizer, Multi-layer perceptron, Wooded environment

Funding: Coordenação de Aperfeiçoamento de Pessoal de Nível Superior-Brasil (CAPES) 001 This study was financed by the Coordenação de Aperfeiçoamento de Pessoal de Nível Superior-Brasil (CAPES)–Finance Code 001. There was no additional external funding received for this study. The funders had no role in study design, data collection and analysis, decision to publish, or preparation of the manuscript.

==============================
The emergence of long-range (LoRa) technology, together with the expansion of uncrewed aerial vehicles (UAVs) use in civil applications have brought significant advances to the Internet of Things (IoT) field. In this way, these technologies are used together in different scenarios, especially when it is necessary to have connectivity in remote and difficult-to-access locations, providing coverage and monitoring of greater areas. In this sense, this article seeks to determine the best positioning for the LoRa gateway coupled to the drone and the optimal spreading factor (SF) for signal transmission in a LoRa network, aiming to improve the connected devices signal-to-noise ratio (SNR), considering a suburban and densely wooded environment. Then, multi-layer perceptron (MLP) networks and generalized regression neural networks (GRNN) were trained to predict the signal behavior and determine the best network to represent this behavior. The MLP network presented the lowest RMSE, 2.41 dB, and was selected for use jointly with the bioinspired Grey-Wolf optimizer (GWO). The optimizer proved its effectiviness being able to adjust the number of UAVs used to obtain 100% coverage and determine the best SF used by the endnodes, guaranteeing a higher transmission rate and lower energy consumption.

Introduction

Long-range (LoRa) technology is expanding, the number of LoRa devices deployed around the world grew from 97 million to 300 million in the period from April 2019 to March 2023 (ABIresearch, 2019; SEMTECH, 2023). This technology uses spread spectrum modulation, enabling operation below the noise floor. That is, considering a fixed signal-to-noise ratio (SNR), only the bandwidth needs to be increased for error-free transmission (Lathi & Ding, 2009). Additionally, the technology uses the spread factor (SF) responsible for adjusting the data transmission rate, range, and consequently energy consumption. Based on this, LoRa technology provides low energy consumption and a long transmission range, making it ideal for Internet of Things (IoT) applications that require device communication over large areas.

Diverse applications use LoRa devices, for example, asset management, environmental monitoring, smart cities, smart agriculture, surveillance, security, and healthcare (LoRa Alliance & Wireless Broadband Alliance, 2020). This wide range of applications and the growth of LoRa technology has caught the interest of researchers, and the technology is currently present in the most diverse research (Seye et al., 2018; Park et al., 2018; Bocker et al., 2019; Chall, Lahoud & Helou, 2019; Mod Rofi et al., 2021).

Furthermore, intending to expand the reach of LoRa communication and reach remote regions that are difficult to access, researchers integrated LoRa technology into uncrewed aerial vehicles (UAVs), also called drones (Dambal et al., 2019; Marchese, Moheddine & Patrone, 2019; Moheddine, Patrone & Marchese, 2019). Drones are frequently employed in mapping missions, industrial inspections, and search and rescue operations. In addition, they have gained ground in civil applications such as agriculture, transport of small loads, security, media, and, more recently, telecommunications (Ghamari et al., 2022).

Finally, advances in drone technologies have also made it possible to use them in the field of wireless communications. Drones act as aerial base stations and have an advantage over terrestrial base stations, they are capable of operate in the most diverse scenarios. Therefore, it is necessary to adequately plan these devices positioning, aiming to improve the performance of the communication network and optimize the coverage area (De Rango & Stumpo, 2023).

It is crucial to consider the region’s unique characteristics when planning a communication network implementation. In this context, the present study directs its analysis to the Federal University of Pará (UFPA) Belém campus, an environment with suburban characteristics and densely wooded, located in Belém-Pa, Brazil. This choice is based on the expectation that wooded areas will expand in cities around the world.

The expansion projection occurs due to government actions around the world, such as the Decade of Ecosystem Recovery (2021–2030) announced by the United Nations (UN) (Zandonai, 2021), which tries to respond to the worsening of the climate crisis. As well as, the Green New Deal, representing public policies that promote economic growth in harmony with more sustainable practices (Alvares, Rodrigues & Narita, 2021). Therefore, when considering a suburban and densely forested scenario, this study considers both the particularities of the environment and global trends in sustainable development.

Given this perspective, this study employs the grey-wolf bioinspired optimization algorithm (GWO), adapted to the peculiarities of UFPA, to determine the optimal number of aerial LoRa gateways and optimize their arrangement to maximize the SNR, a classical measure of quality in both analog and digital systems, reflecting the transmission channel’s and receiver’s performance (Lathi & Ding, 2009; Haykin & Moher, 2006). Ensuring a high SNR helps maintain a clear and strong signal, thereby minimizing the likelihood of errors in data transmission. This is particularly important for applications such as wildlife monitoring, environmental data collection, and communication systems where precise data is essential.

The study goal is to ensure connectivity for the largest possible number of devices within the communication network with the minimum number of UAVs. To achieve this, the GWO is used in conjunction with a trained artificial neural network (ANN) that predicts signal behavior in this environment, enhancing the optimization of the LoRa network. Additionally, the study involves mapping wooded regions and buildings within the UFPA perimeter, allowing the GWO to determine the most appropriate spreading factor (SF) for devices connected to aerial gateways, thereby optimizing message transmission.

The remainder of this article is organized as follows: “Related Works” presents related research, “Methodology” presents the methodology used to model the problem, “Results” presents the results obtained from the execution of the GWO algorithm and “Conclusion” presents the conclusion.

Related works

In the work of Mozaffari et al. (2019), a detailed study is conducted on the use of drones in wireless communications, with emphasis on their application as base stations. Furthermore, the researchers highlight the challenge of configuring aerial base stations considering the three axes of drone movement. This challenge is due to the great mobility of these vehicles, which causes constant changes in the properties of signal’s propagation medium. In addition, the authors emphasize the need for researches that seeks coverage area maximization while recognizing obstacles for aerial base station implementations.

Ghazali, Teoh & Rahiman (2021) proposed to carry out a systematic review of the LoRa communication networks implementation using drones given the current relevance of both technologies for emerging IoT applications. The researchers show several studies that use the drone sometimes as endnode, sometimes as gateway to answer questions about the tendency to use LoRa-Drone technologies together, the reach of the LoRa-Drone network and the performance of this network analyzed based on the packet loss rate. However, the authors focus their research on the LoRa and Drone integration and do not address the issue of optimizing the use of both to improve the coverage area and network performance.

Comprehensive research related to emerging communication solutions for IoT applications is carried out by Vaezi et al. (2022). One of the highlighted topics is the integration of drones into IoT networks. The authors demonstrate that drones have the ability to improve real-time communication by acting as relays between IoT nodes and base stations (BS), resulting in more reliable links, expansion of the coverage area, and the possibility for IoT nodes to reduce transmission power, resulting in lower energy consumption. From the above, there is a need to integrate these technologies and conduct research that allows both to operate at maximum performance.

In studies conducted by Oh, Lim & Kang (2020), Okuda et al. (2022) and Al-Shareeda, Manickam & Saare (2022); integration occurs between the LoRa and UAV technologies with different purposes. In the first work, the authors address the necessity of establishing procedures to identify UAVs and analyze different communication methods to derive the probability of success in identifying a drone, based on LoRa technology. In the second study, the propagation characteristics are examined considering the frequency of 920 MHz applied for the inspection of transmission lines using UAVs. In the last article, the authors propose an intelligent system for farms. The system uses drones with Zigbee and LoRa to collect images of the soil and data from sensors present in the plantation.

A system for precision agriculture is also proposed in Holtorf et al. (2023). The authors use a drone together with LoRa technology to collect information from sensors positioned inside the soil. The study shows that drone usage increased the coverage area from 300 to 1,000 m. Furthermore, the transmitted signal was characterized considering the drone’s height and the node’s depth. Finally, based on the study, authors cite the determination of the ideal trajectory for the drone and optimization of the antenna that transmits from the ground to the air as future work.

Wilson et al. (2022) provides a comprehensive review of the embedded sensors, communication technologies, computing platforms, and machine learning techniques use in UAVs. The study discusses the various sensors and communication technologies used in UAVs, as well as the operational principles and a comparative analysis of these technologies. The article’s purpose is to serve as a reference guide for the design of intelligent sensing applications, low-latency and energy-efficient communication strategies, energy-efficient computing modules, and machine-learning algorithms for autonomous UAVs. Additionally, the authors present open questions and challenges for future research in this field of study.

Abbas et al. (2021) optimize the number of UAVs and their positioning, also decide on the association of IoT devices with UAVs and the allocation of radio resources, enabling the association between devices IoT to several UAVs through orthogonal frequency division multiple access (OFDMA). To achieve this, the authors use a cluster-based approach, which offers low-complexity sub-optimal solutions that demonstrate the relevance of using multiple paths to ensure reliability with minimal resource consumption.

The research carried out by Cruz et al. (2022) proposes a methodology based on measurements to adjust the coefficients of UFPA propagation model for LoRa technology through the application of a genetic algorithm. Furthermore, the authors use the Evolutionary Particle Swarm Optimization (EPSO) algorithm to maximize the coverage area of the LoRa network, with the smallest number of gateways possible. However, the analysis does not consider aerial gateways usage.

Farhad & Pyun (2023) carry out a survey on the use of machine learning (ML) to improve the performance of the Long Range Wide Area Network (LoRaWAN) protocol in the context of IoT. Resource management issues in LoRaWAN and how ML methods can improve network performance are discussed. The study also explores available tools and datasets for managing LoRaWAN resources and evaluates ML tools to manage resources efficiently. Finally, the article presents open problems, directing researchers to define the node’s optimal SF and ideal transmission power.

In the work in Al-Gumaei et al. (2022), the authors introduce BE-LoRa, a tool for LoRaWAN based on game theory. This model seeks to optimize the packet delivery rate and energy efficiency (bit/Joule). LoRa nodes are rational agents that seek to maximize their utility function while maintaining a constant signal-to-interference-to-noise ratio (SINR) for each spreading factor (SF). The results indicate that BE-LoRa performs better than LoRaWAN ADR in terms of packet delivery and energy efficiency. This work suggests the integration of an admission algorithm into BE-LoRa to regulate the number of nodes in each SF as future work.

Saluja et al. (2021) address the scalability of LoRawAN networks and propose an SNR-based SF allocation scheme to improve it. In the study, existing allocation schemes are compared to two proposed algorithms. The authors applied the proposal in a real scenario, and the results obtained show the superior performance of the technique. However, the authors used a terrestrial gateway for the analysis, which performs differently than an aerial gateway.

The works cited above deeply explore the adoption of UAVs in IoT communication networks, highlighting an emerging trend capable of bringing significant changes in a variety of sectors, from agriculture to logistics and environmental monitoring. These works frequently mention LoRa technology, emphasizing the combination between LoRa and UAVs. This integration is a promising solution to optimize and expand the reach and efficiency of LoRa networks, especially in remote or difficult-to-access areas.

From the above, it is possible to observe the existence of several related works that approach the themes of this study. There are studies in which the integration of LoRa and UAV technologies occurs, studies that address the optimal allocation of SF, and studies that deal with the ideal positioning of drones acting as gateways. However, the studies work on the themes separately. In this research, the integration between LoRa and drones is carried out in an experiment to collect real data. Then, from the collected data, two neural network models are trained to reproduce the behavior of the signal. Finally, using the GWO algorithm with help from the trained neural network, the best SF for each device is determined, and the drone positioning is optimized.

According to what was discussed in this session, the motivation for this study lies in the current nature of the topic and the need for planning methods for LoRa networks enabled by drones that consider the particularities of the environment in which the network will be deployed in this case a suburban and densely wooded environment. In this way, the study covers the current topic, LoRa and UAV integration, as well as the global perspectives of sustainable development that encourage cities with more green zones. Table 1 presents the related works contributions summary.

Table 1 Summary of analyzed related work contributions.

Ref.	Contributions	
Mozaffari et al. (2019)	Survey about UAVs as basestations, indicates the need of more researches in UAV positioning optimization field	
Ghazali, Teoh & Rahiman (2021)	Survey about Real-time deployments of UAV-Based LoRa communication networks, focus on researches that integrates UAV and LoRa technologies.	
Vaezi et al. (2022)	Authors show the need of more research focusing on optimization of joint performance for IoT and UAV technologies.	
Oh, Lim & Kang (2020)	Authors integrate LoRa and UAV technologies to derive the UAV identification success probability through various propagation models.	
Okuda et al. (2022)	Authors make the analysis of LoRa 920 MHz propagation characteristics for power transmission lines inspection using LoRa and UAV.	
Al-Shareeda, Manickam & Saare (2022)	Authors integrate LoRa, ZigBee, and UAV to propose a system that helps new farms and farms doing crop rotation.	
Holtorf et al. (2023)	Authors deploy underground sensors and use UAV with LoRa to increase signal range, also evaluate the deploy depth impact in the signal.	
Wilson et al. (2022)	A thorough survey that presents open problems and future challenges related to UAV in diverse study fields, such as, efficient computational modules and machine learing algorithms for autonomous UAVs.	
Abbas et al. (2021)	Optimization of UAV positioning, IoT devices to UAV association and radio resources allocation, through a clustering-based approach.	
Cruz et al. (2022)	UFPA model coefficients adjustment for LoRa technology through genetic algorithm and EPSO to maximize LoRa network coverage area.	
Farhad & Pyun (2023)	Survey about ML applied to LoRaWAN protocol optimization, shows open problems related to SF selection and ideal transmission power.	
Al-Gumaei et al. (2022)	Jointly optimize the packet delivery ratio and the energy efficiency through a game-theoretic framework.	
Saluja et al. (2021)	SF allocation scheme through expressions for packet success probability under co-SF and inter-SF interference scenarios.	

Methodology

This section describes the methodology adopted in this work, the study scenario, data collection, training of neural network models, and modeling of the GWO algorithm. Figure 1 illustrates the methodology flowchart until results are obtained.

Figure 1 Methodology fluxogram.

Study environment

The Federal University of Pará (UFPA) Belém Campus, one of the most important higher education institutions in the North of Brazil, was chosen to carry out the analysis. The campus has approximately 450 hectares located on the Guamá River banks. UFPA has a variety of buildings and wooded regions, offering a diverse environment with practical implications for research related to the broadest themes (Federal University of Pará (UFPA), 2023). Figure 2 illustrates the UFPA Belém Campus geographic location and its perimeter outlined in red.

Figure 2 UFPA (Map data ©2023 Google).

Data collect

In this study, the data presented in Cardoso et al. (2022) was used. An Inspire One drone, two LoRa Dragino with GPS, and a car were used to collect data. The data was transmitted from a LoRa Dragino attached to the car and received on a LoRa Dragino attached to the drone. The transmission frequency was 915 MHz; the established drone’s heights were 6, 24, 42, and 60 m; SFs considered for analysis were 8, 9, 10, and 11; the transmission used 20 dBm of power and 500 KHz of bandwidth; the computer located in the car stored the data. Figure 3 illustrates the drone and radio used.

Figure 3 Measuring equipment.

The car went through two distinct routes to collect data. The first route, known as the “garage route” encountered numerous obstructions within the initial meters. In contrast, the second route, referred to as the “ceamazon route” experienced minimal obstructions within the initial meters. The raw data is provided in a CSV file containing the following columns: drone’s height, drone’s geographical coordinates, SF, Bandwidth, car’s geographical coordinates, RSSI, SNR, distance, radial distance, received power, a flag indicating whether the route is ceamazon (0) or garage (1), and a flag indicating whether the transmission is downlink (1) or uplink (0).

The total number of samples after measurements was 3,614 samples, and the SNR values collected were in the range of −18 to 9 dB, considering samples from downlink, drone-car communication, and uplink, car-drone communication. The variation of SF and drone height directly impacted the maximum communication range. Higher values provided a greater range.

For downlink, the range was 1.1 km, and in uplink communication, the maximum range was 0.5 km. It is believed that range difference occurs due to the difference in the propagation medium during signal formation, as in downlink, the electromagnetic wave is formed in the air without obstructions. While in uplink, the wave is formed in a terrestrial environment obstructed by buildings and wooded regions. Table 2 exihibts part of the processed data used in the neural network model.

Table 2 Processed data.

Distance (m)	SF	Height (m)	Uplink = 1 or Downlink = 0	SNR (dB)	
364.93	9	60	0	7	
104.84	8	6	0	3	
332.22	10	42	0	−4	
375.97	11	24	1	0	
680.98	11	42	1	0	
687.27	8	24	1	−4	
888.79	10	6	0	5	
98.02	11	24	1	6	
225.15	8	6	0	3	
.	.	.	.	.	
.	.	.	.	.	
.	.	.	.	.	

Neural model training

Several multi-layer perceptron (MLP) and general regression neural networks (GRNN) were trained using the collected data, considering different input parameters. To determine the best inputs and the best model, four GRNNs and four MLPs were modeled based on the combination of input parameters: distance (d), height (h), SF, free space path loss (FSPL) and flag uplink-downlink (UD), used to distinguish between car-drone and drone-car communication.

For MLP-type networks, the model chosen had a hidden layer and an output layer, whose activation functions are logsig and purelin, respectively. Furthermore, the number of neurons in the hidden layer varied from one to 20, resulting in 80 MLP models, one for each combination of input parameters and number of neurons.

For GRNN-type networks, the architecture had a hidden layer with numbers of neurons equal to the number of samples used in network training, a Gaussian activation function, and a neuron in the output layer with a purelin activation function.

Additionally, during the training process, the cross-validation technique with k=10 folds was used (Gholamiangonabadi, Kiselov & Grolinger, 2020). In this way, ten training sessions will be performed for each of the 80 MLP models and 4 GRNN models, generating 800 MLP models and 40 GRNN models. Furthermore, to generate statistical validation, this training process is repeated 100 times (Jain, 1991), generating a total of 84,000 neural network models. Figure 4 illustrates the boxplot of the root mean squared error (RMSE) for all models trained considering different combinations of inputs.

Figure 4 Comparison between MLP × GRNN topologies.

Figure 4 illustrates the RMSE for the four combinations of input parameters. Through its analysis, it is possible to observe that there is not a big difference between the MLP and GRNN models, the difference is less than 1 dB. Regarding the input parameters, there is a high standard deviation and several outliers when using only the distance and flag UD as network input. Note that when using three parameters, there is also a large presence of outliers. The result is very similar for four and five parameters, but the number of outliers is smaller. Therefore, it is observed that, in this case, the addition of input parameters reduced the variability of the results presented by the different neural models.

From the analysis above, it was determined that models 3 and 4 exhibited similar performance. For the final model selection, model 3, with four input parameters distance, height, SF, and flag UD was chosen due to its lower RMSE variability and reduced computational cost compared to the model with five input parameters. To determine the optimal number of neurons in the hidden layer, 100 simulations were performed for each topology, and outliers were removed using a 2.5 median absolute deviation (MAD) criterion. Among the remaining neural network models, the MLP network with 17 neurons in the hidden layer presented the lowest RMSE of 2.41 dB and was selected for use alongside the GWO algorithm. Figure 5 illustrates the topology of the selected network.

Figure 5 MLP network topology.

Grey wolf optimizer

The grey wolf algorithm (GWO) (Mirjalili, Mirjalili & Lewis, 2014) copies the hunting mechanism and leadership hierarchy of these wolves to identify optimal solutions to problems in general. For the GWO to work, it is necessary to implement the hierarchy of alpha, beta, delta, and omega wolves, as well as the three main stages of the hunt: searching for prey, surrounding prey, and attacking prey.

Wolf hierarchy

Wolves are classified into four categories: Alpha wolf ( α) is the leader of the pack, the dominant wolf, and the best solution to the problem; Beta wolf ( β) is at the second level of the pack, subordinate to α, assists in the group’s decision making and is the second best solution to the problem; Delta wolf ( δ) must obey α and β acts as scout and sentry for the pack, it is the third best solution to the problem; Omega wolf ( ω) must submit to the others, and is the scapegoat serving for the other wolves to relieve their stress and maintain the pack hierarchy, for the algorithm the wolves ω are the other candidate solutions.

Search and attack prey

Grey wolves search for prey according to the positioning of wolves α, β and δ. These wolves walk in opposite directions while searching for prey, then converge to attack. This behavior is modeled in Eq. (1), as shown in

(1) A→=2a→⋅r1→−a→

where A→ is a coefficient vector, a→ linearly decreases from 2 to 0 over the iterations, and r1→ is a random value in the interval [0,1]. While A→ is greater than 1, the wolves spread throughout the search space. When A→ is less than or equal to 1, the wolves converge to attack the prey.

Surround the prey

When grey wolves find prey, they surround it. To represent this behavior Eqs. (2)–(4) are used.

(2) C→=2⋅r2→

(3) D→=|C→⋅Xp→(t)−X→(t)|

(4) X→(t+1)=Xp→(t)−A→⋅D→

where C→ is a coefficient calculated from the random value r2 in the interval [0,1], Xp→ is the position of the prey vector, and X→(t) is the position of the grey wolf vector.

Hunt the prey

The hunt is guided by the α wolf, considered the best candidate solution, followed by the β and δ wolves. Assuming that these wolves have greater knowledge about the possible location of the prey, the three positions are stored, and all wolves move based on them. Equations (5)–(7) were modeled to update the position of any wolf X→(t), and are described by

(5) D→α=|C→1⋅X→α−X→(t)|,D→β=|C→1⋅X→β−X→(t)|,D→δ=|C→1⋅X→δ−X→(t)|,

(6) X→1=X→α−A→1⋅D→α,X→2=X→β−A→2⋅D→β,X→3=X→δ−A→3⋅D→δ,

(7) X→(t+1)=X→1+X→2+X→33,

where X→1 represents the update factor related to component α, X→2 represents the update factor related to component β, X→3 represents the update factor related to the δ component and X→(t+1) is the new position of the grey wolf.

Grey wolf optimizer modeling

To model the problem, the devices used in the Amazon Multimodal Intelligent System (SIMA) (Lobato et al., 2023) project were considered. In this project there are seven endnodes, one for the photovoltaic system, one for the battery system, two divided for two electric stations, two divided for two electric buses, and one for an electric boat. Figure 6 illustrates the scenario with the presence of seven endnodes, of which four are immobile, represented by purple circles, and three are mobile, represented by yellow and green diamonds.

Figure 6 Initial scenario (Map data ©2023 Google).

Furthermore, it was necessary to map the wooded regions and building regions of UFPA, for which the Google Maps API V3 Tool (https://www.birdtheme.org/useful/v3tool.html) was used. This way, it is possible to ensure that according to the geographic coordinate, the optimizer knows the correct height to position the drones. Figure 7 illustrates the mapping carried out, the wooded regions in green and the building regions in light blue.

Figure 7 Trees and buildings mapping (Map data ©2023 Google).

From the information above, it is possible to model the grey wolf for the desired problem. The wolf is composed of the used drones’ number, their heights, their geographic coordinates, the SF used to communicate with each device, the SNR of each drone in relation to each device, and its fitness. Figure 8 illustrates the grey wolf vector modeled for the problem.

Figure 8 Grey wolf vector.

The problem fitness function is defined by two objectives: the first is to maximize the average SNR of all devices present in the IoT network, and the second is to use the smallest number of drones possible to offer complete coverage. In this way, a table is created containing the SNR values of each endnode in relation to each drone. Table 3 illustrates an example for the case of three drones (D) and seven endnodes (E).

Table 3 Example of the SNR (dB) drone × Endnode table.

	E1	E2	E3	E4	E5	E6	E7	
D1	3	2	3	1	3	4	1	
D2	5	1	2	4	2	1	3	
D3	2	4	6	3	1	2	4	

From Table 3, it is possible to determine objective 1, maximizing the SNR of LoRa network devices; for this, Eq. (8) is used:

(8) Obj1=mean(max(Ci))SNRmax

where Ci is the i-th column of the matrix and SNRmax is the highest SNR value present in the collected data. To determine the second objective, minimizing the number of drones in use, Eq. (9) is used:

(9) Obj2=1−(NDloboNDmax)

where NDlobo is the drone number in the grey wolf whose fitness is being calculated, and NDmax is the maximum number of drones allowed by the optimizer. Finally, fitness is determined by Eq. (10):

(10) fitness=−[W⋅Obj1]−[(1−W)⋅Obj2]

where W is the Pareto front used to calculate fitness.

Restrictions

The pack initialization needs to respect the restrictions defined for this problem. There are restrictions related to endnodes and restrictions associated with drones. Each endnode sends different information, so they have distinct payloads. Since the SF used limits the LoRa transmission rate, the device payload defines the threshold of SFs used in transmission. Table 4 illustrates the payload of all endnodes and the minimum and maximum SF thresholds according to the AU915 region (LoRa Alliance, 2022).

Table 4 Endnodes payloads and SFs.

Application	Payload (Bytes)	SF range	
Gym electric station	125	8	
Ceamazon electric station	125	8	
Photovoltaic system	53	8–9	
Battery system	53	8–9	
Electric bus 1	23	8–11	
Electric bus 2	23	8–11	
Electric boat	23	8–11	

For drones, they must respect the UFPA limits presented in Fig. 2, that is, the optimizer must position the drones within the perimeter belonging to UFPA. If they are outside, the optimizer repositions them to the closest position within the perimeter. Furthermore, the positioning of the drones must respect the height of the mapped buildings and trees, represented in Fig. 7. For this purpose, the average height of 12 m is considered for all building regions and 25 m for the wooded. The Algorithm 1 exemplifies the adapted grey wolf optimizer pseudocode and the Fig. 9 illustrates the optimization fluxogram.

Algorithm 1 Grey wolf optimizer adapted algorithm.

n= number of search agents	
Input=[nDrones,Height,Lat,Lon,SF,SNR]	
Initialize the Wolf Pack (IWP) of n wolves randomly	
  for i=1:n do	
       Input=random(6,1)	
       IWPi=Input	
  end for	
IWP = Check Restrictions(IWP)	
  for Pareto=1:Paretomax do	
       WolfPack=IWP	
       W=ParetoParetomax	
     while iter≤Itermax do	
      Calculate fitness Fi(i=1,...,n) for all WolfPack (Eq. (10))	
      Sort WolfPack according to Fi in ascending order	
       Xα=WolfPack[1], Best solution; Xβ=WolfPack[2], 2a best; Xδ=WolfPack[3], 3a best;	
      Updates wolves in WolfPack	
        for j=1:n do	
           WolfPack[j]=Xα+Xβ+Xδ3 (Eq. (7))	
        end for	
       WolfPack= Check Restrictions( WolfPack)	
      Decreases Linearly A (Eq. (1))	
      Update C (Eq. (2))	
      iter++	
     end while	
       Solutionpareto=Xα	
  end for	
Output = Best(Solutionpareto)	

Figure 9 Optimization fluxogram.

Optimization problem

Let’s consider the problem above, let n be the number of endnodes in the network. Each endnode is positioned in a Lati and Longi for i=1,...,n. Also, each endnode has a maximum SFi between 8 to 11, and each endnode i has a SNRi,j related to k LoRa Gateways (LG) or drones for k=1,...,j.

The objective function defined in Eq. (10) has the goal to maximize the network SNR with the smallest number of drones possible. For this, j drones will be deployed for j=1,...,k and each drone is positioned at a Latj, Longj and height ( hj) for j=1,...,k. The table SNRi,j represents the SNR from endnode i related to LG j. The SNRi,j is determined through the ANN, Eq. (11), considering the radial distance dri,j between the LG j and the endnode i, the SFi and the LG j height.

(11)  SNRi,j=net(dri,j,hj,SFi,UD)

and dri,j is determined through the adapted Haversine equation, Eq. (12),

(12) dri,j=(2rarcsin⁡(sin2(Latj−Lati2)+cos⁡(Lati)cos⁡(Latj)sin2(Longj−Longi2)))2+hj2,

where dri,j is the radial distance between the LG j and the endnode i; r is the earth radius in meters (6,371,000 m). To summarize, the optimzation problem is modeled as:

minimize Eq. (10)

subject to

 8≤SFi≤11fori=1,…,n−1.4784≤lati≤−1.4584fori=1,…,n−48.4594≤longi≤−48.4346fori=1,…,n−1.4784≤latj≤−1.4584forj=1,…,k−48.4594≤longj≤−48.4346forj=1,…,k6≤hj≤60forj=1,…,kifTreeRegionthenhj≥25ifBuildingRegionthenhj≥12

Results

This section will present the results obtained in this study, the behavior presented by the proposed prediction model, the convergence curves of the optimization algorithm used, and the performance of the optimizer with the analysis of coverage, number of drones, variation of SF and SNR before and after the optimization process.

Neural network

The MLP neural network with 17 neurons in the hidden layer was selected for use in conjunction with the grey wolf optimizer as it presented the best performance. Next, Fig. 10 compares the predicted SNR behavior for all SFs of the selected MLP network, considering the height of 24 m, to the measured data. The chosen network was able to follow the data trend and presented the expected behavior, that is, the reduction in the predicted SNR value occurs as the distance between the transmitter and receiver increases.

Figure 10 Prediction model developed.

Optimizer

From the initial scenario, presented in Fig. 6, 30 search agents are defined randomly, they are organized in ascending order, and their position in the rank may vary according to the Pareto front considered. For the analysis of initial Pareto fronts, fitness prioritizes reducing the number of drones, as seen in Fig. 11. Initially, the optimal solution had only one UAV, illustrated by the green diamond. After the optimization, only one UAV was used in the final solution, represented by the yellow diamond.

Figure 11 Pareto 1 positioning (Map data ©2023 Google).

The final Pareto fronts prioritize maximizing the SNR of devices connected to the LoRa network regardless of the number of UAVs used, as seen in Fig. 12, which illustrates the optimization result for Pareto Front 10. The initial solution had 7 UAVs, and at the end of the optimization process, the use of 8 UAVs was presented as the optimal solution.

Figure 12 Pareto 10 positioning (Map data ©2023 Google).

Next, Fig. 13 illustrates the convergence of the GWO algorithm for the two cases mentioned above. The lower the fitness, the better the individual. There is a considerable difference between the two scenarios caused by the increase in the number of UAVs at the highest Paretos, increasing the complexity of the problem to be solved. The complexity can be seen in the high population diversity present in Fig. 13B compared to that presented in Fig. 13A. For the other Pareto fronts, the algorithm convergence can be seen in Supplemental Files.

Figure 13 Pareto 1 & 10 convergence.

For a general analysis of all Pareto fronts, Fig. 14 is presented. In blue are the initial values, and in red are the post-optimization values. From the figure analysis, it is possible to observe that, for Paretos 1 to 6, only 1 UAV is used at the end of the optimization. However, this amount is insufficient to provide coverage for the entire LoRa network. In this way, it is noted that the most appropriate Pareto for solving the problem is between 7 and 10 since these are the ones that provide full coverage for the communication network.

Figure 14 Analysis of pareto fronts.

Therefore, restricting the analysis to Paretos between 7 and 10, it is observed that Paretos 7 and 8 achieved the objective of reducing the UAVs number and improving the SNR of devices connected to the network, placing UAVs at 46 and 52 m height, respectively. Both Paretos achieved 100% coverage for the LoRa network, reducing the number of UAVs used from 3 to 2 and 7 to 2, respectively. Furthermore, Paretos 7 and 8 were able to reduce the average SF used, providing a higher transmission rate for endnodes and reducing energy consumption during data transmission. Finally, Fig. 15 presents the positioning for Paretos 7 and 8, and the others can be seen in Supplemental Files.

Figure 15 Pareto 7 & 8 positioning.

Conclusion

This study proposed the grey wolf optimizer utilization to determine the optimal positioning of UAVs in a densely wooded suburban region and select the appropriate spreading factor for data transmission from endnodes connected to the LoRa network. To this end, the mapping of buildings and trees within the UFPA perimeter was carried out. Additionally, samples of the LoRa signal behavior were collected in this environment, and several neural networks were trained to reproduce this behavior, aiming to assist the LoRaWAN network planning process.

In total, 84,000 neural networks were trained. After training, the perceptron multi-layer network with 17 neurons in the hidden layer presented the lowest RMSE, 2.41 dB. This network was used in conjunction with the optimizer, which considered the positioning of the endnodes used by the SIMA project to determine the optimal position of the UAV gateways. Furthermore, restrictions were defined so that the UAVs are positioned respecting the heights of buildings and trees and restrictions so that the SF selected for transmission is in accordance with the payload used by the endnode.

The results show that the optimizer had acceptable performance, achieving 100% coverage for devices connected to the LoRaWAN network. Furthermore, the optimizer caused the devices to carry out their transmissions with lower SF, resulting in a higher transmission rate and lower energy consumption. Finally, the methodology presented in this work is expected to assist in future deployments of LoRa networks that use UAVs as gateways.

The plan for future work is to implement the automatic detection of wooded areas and buildings through computational intelligence techniques. The goal is to model a convolutional neural network to determine the different types of terrain and structures from aerial imagery. Also, we plan to use this study to adjust the positioning of UAVs in real-time by applying mobility simulation models to the network users, such as the Random Waypoint.

Supplemental Information

Supplemental Information 1 Dataset and codes used in the research.

The code (otimizador_ufpa) to run the adapted gwo optimizer for the UFPA scenario.

- code "main.m" runs the optimizer

- code "trata_resultados.m" plots the general analysis for all paretos

- folder "resultados" contains the plots used in the draft and the results

- obtained for the optimizer in .mat files

The code (SNR_mlp) to train the grnn and mlp networks:

- "old_mlp_snr_v2.m" trains the mlp networks

- "grnn_snr.m" trains the grnn networks

- "analise_resultados.m" generates the boxplot to compare grnn x mlp, then find the best network

Supplemental Information 2 Measured RSSI and SNR.

Supplemental Information 3 GWO convergence for Pareto 2.

Supplemental Information 4 GWO convergence for Pareto 3.

Supplemental Information 5 GWO convergence for Pareto 4.

Supplemental Information 6 GWO convergence for Pareto 5.

Supplemental Information 7 GWO convergence for Pareto 6.

Supplemental Information 8 GWO convergence for Pareto 7.

Supplemental Information 9 GWO convergence for Pareto 8.

Supplemental Information 10 GWO convergence for Pareto 9.

Supplemental Information 11 GWO final positioning for Pareto 2.

Supplemental Information 12 GWO final positioning for Pareto 3.

Supplemental Information 13 GWO final positioning for Pareto 4.

Supplemental Information 14 GWO final positioning for Pareto 5.

Supplemental Information 15 GWO final positioning for Pareto 6.

Supplemental Information 16 GWO final positioning for Pareto 9.

Additional Information and Declarations

Competing Interests

Author Contributions

Data Availability

The authors declare that they have no competing interests.

Caio M. M. Cardoso conceived and designed the experiments, performed the experiments, analyzed the data, performed the computation work, prepared figures and/or tables, authored or reviewed drafts of the article, and approved the final draft.

Alex S. Macedo performed the experiments, analyzed the data, prepared figures and/or tables, and approved the final draft.

Filipe Cavalcanti Fernandes performed the computation work, prepared figures and/or tables, and approved the final draft.

Hugo A. O. Cruz performed the computation work, authored or reviewed drafts of the article, and approved the final draft.

Fabrício J. B. Barros conceived and designed the experiments, authored or reviewed drafts of the article, and approved the final draft.

Jasmine P. L. de Araújo conceived and designed the experiments, authored or reviewed drafts of the article, and approved the final draft.

The following information was supplied regarding data availability:

The data and code are available in the Supplemental File.

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
