# Peer review of "Automatic spread factor and position definition for UAV gateway through computational intelligence approach to maximize signal-to-noise ratio in wooded environments"

_PeerJ Computer Science, doi:10.7717/peerj-cs.2237_

## Round 0.1 · original submission · Major Revisions

This paper needs a revised introduction to clearly outline the unique aspects of the study, strengthen and justify why the methodologies used, and bolster the validation of the results.

Reviewer 1 ·

Basic reporting

All comments have been added in detail to the 4th section called additional comments.

Experimental design

All comments have been added in detail to the 4th section called additional comments.

Validity of the findings

All comments have been added in detail to the 4th section called additional comments.

Additional comments

Review Report for PeerJ Computer Science
(Automatic SF and position definition for UAV gateway through computational intelligence approach to maximize SNR in wooded environments)

1. Within the scope of the study, it was aimed to determine the best position and optimal spreading factor for Unmanned Aerial Vehicles.

2. In the Introduction section, Long-Range technology and the importance of the subject are mentioned in detail and sufficiently.

3. In the Related works section, existing studies in the literature in this field and their main contributions are basically mentioned.

4. The dataset, number of samples and SNR values used in the study are sufficient.

5. Multi-Layer Perceptron Network and General Regression Neural Networks were used in the study to predict signal behavior. Although there are different neural networks that can be used for this process in the literature, it should be explained in more detail why these two models are preferred.

6. For the optimizer, bioinspired Grey-Wolf optimizer was preferred in the study. Although the optimizer chosen to solve the problem is suitable, explain why this optimizer is preferred and the reasons why a different optimizer is not used and/or preferred.

7. The results and RMSE obtained as a result of the study are sufficient.

8. It is recommended that the future work section in the Conclusion section be detailed a little more.

As a result, although the study is sufficient in terms of the problem and originality addressed, it is recommended to examine the parts mentioned above.

Reviewer 2 ·

Basic reporting

The following issues require further consideration:

1. The author needs to explain how the solution proposed in this article is innovative.

2. The author should provide a solid rationale for choosing to maximize SNR in wooded environments, thereby enhancing the reader's comprehension of the author's approach.

3. Figure 1 shows the Methodology Fluxogram but does not reflect logic and interactivity. It is recommended that it be improved.

4. Figures 5 and 6 do not need to be included in the text; they only need to be quoted.

5. The authors should model the optimization problem for a reader to understand its nature.

Experimental design

The authors need to select some representative benchmark methods and provide their sources.

Validity of the findings

The methods and means of comparison are relatively limited.

Reviewer 3 ·

Basic reporting

Keywords in a paper are typically closely related to the research question, methods, results, or applications, reflecting the main content. In the given sentence, terms like "LoRa," "SF" may not effectively convey the paper's central theme. To facilitate readers, it is recommended that authors reevaluate and select keywords, avoiding abbreviations, and opt for alternatives like "multi-layer perceptron," "grey wolf algorithm," or similar terms that better align with the paper's content.

The paper is written in clear, concise, and professional English, but there are some formatting issues. Equation (8), (9), and (10) on page 11 should be typeset with proper alignment. The variable font in the formula on line 282 should be consistent with that used for standalone equations.

Experimental design

I commend the authors for their extensive use of drones for data acquisition, but to enhance the clarity of the raw data provided, I suggest authors detail the types of data collected, their distinctive features, and the classification scheme in Section 3.1.

A good algorithm should have clear inputs and outputs, and the pseudocode for Algorithm 1 is lacking both input and output specifications.

The author should provide a clarification on the source of the 17-neuron multi-layer perceptron in the text, explaining the rationale behind the chosen neuron count and its claimed optimal performance.

The Grey Wolf Optimizer algorithm is known for its premature convergence issue and tendency to get stuck in local optima. The authors address these drawbacks by outlining their approach to avoid these pitfalls and successfully achieve the global optimum.

Validity of the findings

The paper's experimental results are comprehensive and effectively substantiate the presented arguments.

---

## Round 0.2 · accepted · Accept

Authors have addressed all the comments from the reviewers. Therefore, this paper is recommended to be accepted in its current form.

Reviewer 1 ·

Basic reporting

All comments have been added in detail to the last section.

Experimental design

All comments have been added in detail to the last section.

Validity of the findings

All comments have been added in detail to the last section.

Additional comments

Review Report for PeerJ Computer Science
(Automatic Spread Factor and position definition for UAV gateway through computational intelligence approach to maximize SNR in wooded environments)

Thanks for the revision. It is observed that all revisions made to the paper are sufficient. I recommend that the paper be accepted due to both its contribution to the literature and its originality. I wish the authors success in their future studies. Best regards.

Reviewer 3 ·

Basic reporting

After the author's revisions, the paper is clear and logical in writing.

Experimental design

Research question well defined, relevant & meaningful. It is stated how research fills an identified knowledge gap.

Validity of the findings

All underlying data have been provided; they are robust, statistically sound, & controlled.